# Review of Natural Compounds for the Management and Prevention of Lymphoma

**Yongmin Cho** [1,2,†], **Moon Nyeo Park** [1,2,†], **Seungjin Noh** [3], **Seog Young Kang** [3] and **Bonglee Kim** [1,2,3,*]

[1] Department of Pathology, College of Korean Medicine, Graduate School, Kyung Hee University, 1 Hoegi-dong, Dongdaemun-gu, Seoul 130-701, Korea; ymcho@khu.ac.kr (Y.C.); mnpark@khu.ac.kr (M.N.P.)
[2] Korean Medicine-Based Drug Repositioning Cancer Research Center, College of Korean Medicine, Kyung Hee University, Hoegi-dong, Dongdaemun-gu, Seoul 05253, Korea
[3] College of Korean Medicine, Kyung Hee University, 1 Hoegi-dong, Dongdaemun-gu, Seoul 130-701, Korea; nohapril@khu.ac.kr (S.N.); pionasy@khu.ac.kr (S.Y.K.)
[*] Correspondence: bongleekim@khu.ac.kr; Tel.: +82-2-961-9217
[†] These authors equally contributed to this study.

**Abstract:** Lymphoma is a type of blood cancer that can be categorized into two types-Hodgkin lymphoma (HL) and Non-Hodgkin lymphoma (NHL). A total of 509,590 and 79,990 cases of NHL and HL were newly diagnosed in 2018, respectively. Although conventional therapy has stridden forward over recent decades, its adverse effects are still a hurdle to be solved. Thus, to help researchers develop better lymphoma treatment, this study aims to review the systematic anticancer data for natural products and their compounds. A variety of natural products showed anticancerous effects on lymphoma by regulation of intracellular mechanisms including apoptosis as well as cell cycle arrest. As these results shed light on the potential to substitute conventional therapy with natural products, it may become a promising strategy for lymphoma treatment in the near future.

**Keywords:** lymphoma; cancer system biology; natural products; apoptosis; cell cycle arrest

## 1. Lymphoma

Cancer is among the leading causes of death, which is affected by many factors such as heredity, environment, and lifestyle. Among cancer, lymphoma is the most common blood cancer. It can be categorized into two types, Hodgkin lymphoma (HL) (10%) and Non-Hodgkin lymphoma (NHL) (90%) based on cell of origin, the normal counterpart. Each lymphoma has a number of subtypes, such as diffuse large B cell lymphoma (DLBCL), follicular lymphoma (FL), and Burkitt's lymphoma (BL). Most of the lymphomas were originated from B cell [1]. According to Global Cancer Observatory, 509,590 and 79,990 cases of NHL and HL were newly diagnosed, respectively, in 2018. HL that is characterized by the existence of Reed-Sternberg cell, which occupies 10% of all lymphomas, while NHL accounts for 90% [2,3]. Lymphomas affect any organs in the human body showing a wide range of symptoms including sweat, fever, weight loss, etc. [4]. Conventional therapies including radiotherapy, chemotherapy, and combination therapy can be recommended and in case of indolent lymphoma, careful observation can be used [1]. Although conventional therapies have stridden forward over recent decades, it is still a noncurable disease, and its treatment has several side effects [5]. This is why new drugs for lymphoma need to be developed soon.

## 2. Cancer System Biology

Cancer is reported to be the leading cause of death for humankind worldwide, and side effects of chemotherapy and relapse are unfavorable and remain unsolved problems [6]. To overcome hostile limitations, we need to fully understand how to characterize cancer. Owing to the human genome project, it is possible to elucidate the specific molecular network with therapeutic potential in anticancer mechanisms. Thus, researchers focused on elucidating single gene function or single mutations [7]. However, cancer is a complex ecosystem where heterogeneous tumor cells and nontumor cells coexist [8]. The purpose of cancer system biology is to develop a holistic view of this complex disease, cancer. Moreover, system biology approaches can be used for effective strategies to personalize cancer therapy. Thus, for development of novel lymphoma treatment, in this study, underlying anticancer mechanisms of natural products were collected and organized for better understanding of this dreadful disease.

## 3. Natural Products and Cancer

To date, it has been reported that natural products derived from microorganisms, plants, and animals provide people with improvement of health condition. Resveratrol, a well-known phytochemical, being a component of various plants including berries, has been reported to have anticancer effects as well as antioxidation properties [9]. Moreover, it has been known that curcumin, which is found in Indian spice, exerts anti-inflammatory and anticancer effects [10]. Our team reported that several natural products showed anticancer effects against hematological cancers. *Spatholobus suberectus* Dunn (SSD) regulated the miR-657ATF-2 and induced endoplasmic reticulum (ER) stress related apoptosis along with reactive oxygen species (ROS) generation [11]. Additionally, SSD has been reported regarding its anticancer effect in breast cancer and glioma [11–14]. To date, *Salvia miltiorrhiza* (SM) is one of the well-known natural components considered as a cure for various human diseases. SM showed an anticancer effect through regulation of miR-216b/c-Jun and ER stress related apoptosis pathways [15]. The anticancer effects of *Cnidium officinale* Makino (COM) interact with the miR-211/CCAAT/enhancer-binding protein homologous protein (CHOP) and ROS induced apoptosis pathway in multiple myeloma and myeloid leukemia [16]. Accumulating evidence showed that the natural products can be developed for cancer clinical applications. Thus, focusing on the potential of lymphoma treatment, we investigated many natural components including resveratrol, curcumin, and various other kinds. Components were classified into three groups by their origins-microorganisms, plants, and animals in accordance with their sources. The main purpose of this study is to provide researchers with systematic data about natural products for lymphoma treatment to inspire the development of novel drugs for lymphoma.

## 4. Methods

To collect studies with respect to natural products for lymphoma, Pubmed (https://pubmed.ncbi.nlm.nih.gov/) was used. The following words "natural product and lymphoma" and "natural compound and lymphoma" were used as keywords. There was a term limit of the last 10 years. Collected studies were subjected to the following criteria: whether (1) studies contain in vitro or in vivo experiments, (2) studies include natural product or compound not synthetic, (3) data presented in studies were statistically reliable (p- values that were less than 0.05). Then, we classified data into three categories in accordance with their sources.

## 5. Natural Compounds and Lymphoma

### 5.1. Microorganism-Derived Compounds and Lymphoma

Microorganisms with a microscopic size are referred to as prokaryotes. Fungi, which are classified as eukaryotes, are also included in microorganisms in a broad sense. In recent decades, it has been reported that microorganism-derived compounds have various bioactivities

including antioxidant and antitumor effects [17,18] (Table 1). According to studies, microorganism derived compounds induced cell cycle arrest at the G2/M phase and inhibited tubulin assembly; subsequently, it triggered apoptosis in colon cancer cells [19]. Several natural compounds from *Aspergillus fumigatus* YK-7 significantly inhibited growth of U937 cells after three days of treatment; the natural compounds included pyripyropene E, alismol, helvolic acid, and β-5, 8, 11-trihydroxybergamot-9-ene [20]. These toxic effects against lymphoma cell lines supported the potential of natural compounds as active agents to treat various types of lymphoma. Bao et al. isolated new compounds from the coculture of marine-derived fungi *Aspergillus sclerotiorum* and *Penicillium citrinum* [21]. Among the compounds, aluminiumneohydroxyaspergillin exhibited cytotoxicity against the U937 cell line (IC50 = 4.2 μM). L. Hammerschmidt et al. isolated twelve compounds from *Gymnascella dankaliensis* ethyl acetate extract and measured their cytotoxicity against the L5178Y cell line [22]. Four compounds were found to have cytotoxicity. Aranorosin-2-methylether, aranorosin, and gymnastatin A showed potent cytotoxic activity against L5178Y cells with IC50 values of 0.44, 0.58, and 0.64 μM, respectively. Gymnastatin B had moderate cytotoxic activity with an IC50 value of 5.8 μM. Three natural products isolated from *Aspergillus carneus*, isopropylchetominine, sterigmatocystin, and astelotoxin E showed significant cytotoxicity against mouse lymphoma cell L5178Y (IC50 0.04, 0.3, and 0.2 μM, respectively) [23]. Bromophilone B isolated from sponge-associated fungus *Penicilliun canescens* displayed cytotoxicity against L5178Y, a mouse lymphoma cell line (IC50 8.9 μM) [24]. *Penicillium citrinum* var. originated natural compounds, including 5-methyl alternariol ether, methyl, 8-hydroxy-6-methyl-9-oxo-9H-xanthene-1-carboxylate, and citriquinochroman, showed toxic events against L5178Y cells each at doses of IC50 0.78, 1.0 μg/mL, IC50 0.78, 1.0 μg/ml, and IC50 6.1 μM [25,26]. Cladosporinone and viriditoxin which are derived from *Cladosporium Cladosporioides* (Fresen). G.A. de Vries also induced cytotoxicity when administered to L5178Y cells in IC50 0.88 and 0.1 μM [27]. Verticillin D and new cyclic heptapeptides Cyclo-(Gly-D-Leu-D-allo-Ile-L-Val-L-Val-D-Trp-β-Ala) isolated from soil-derived fungus *Clonostachys rosea* displayed significant cytotoxicity against L5178Y mouse lymphoma cell lines (IC50 0.1 and 4.1 μM, respectively) [28]. Yu et al. investigated metabolites from ascomycete fungus *Aphanoascus fulvescens* isolated from goose dung [29]. Indole alkaloids, okaramine A, C, G, and H were evaluated to induce cytotoxicity against L5178Y cell lines at different IC50 concentrations, among which okaramine G showed the highest significance. Umeokoli et al. isolated 9 compounds from *Lasiodiplodia theobromae* M4.2–2 [30]. Among the compounds, 1 H-Dibenzo (b,e) (1,4) dioxepin-11-one, 3, 8-dihydroxy-4-(methoxymethyl)-1, 6-dimethyl exhibited potent cytotoxicity against L5178Y mouse lymphoma cell lines (IC50 = 7.3 μM). P.F. Uzor et al. isolated three compounds from the fungus *Nigrospora oryzae* and six compounds from its host plant, *Combretum dolichopetalum* [31]. Two out of these nine compounds were found cytotoxic against L5178Y cells through the cytotoxicity assay (MTT). 4-dehydroxyaltersolanol A from *Nigrospora oryzae* ethyl acetate extract had the IC50 value 9.4 μM, and 3, 3′, 4-tri-O-methylellagic acid from *Combretum dolichopetalum* methanol extract showed the IC50 value of 29.0 μM. 9-Ethyliminomethyl-12-(morpholin-4-ylmethoxy)-5,8,13,16-tetraaza -hexacene-2,3-dicarboxylic acid (EMTAHDCA) isolated from fresh water cyanobacterium *Nostoc* sp. MGL001 showed significant cytotoxicity against the DLA cell, concomitant with 372.4 ng/mL of IC50 value [32]. As shown in Table 1, twenty-seven compounds from microorganism exert cytotoxicity against lymphoma cell lines. Among them, Alismol, Helvoic acid, and β-5,8,11-trihydroxybergemot-9-ene from *Aspergillus fumigatus* YK-7 had IC50 values more than 50 μM, but several compounds, including aranorosin and gymnastatin A from *Gymnascella dankaliensis,* were less than 1 μM [20]. These are somewhat prominent values compared with other values.

**Table 1.** Microorganism-derived compounds and lymphoma.

| Compound | Source | Cell Line/Animal Model | Dose; Duration | Efficacy | Reference |
|---|---|---|---|---|---|
| Alismol | *Aspergillus fumigatus* YK-7 | U937 | IC50 67.1 µM; 3 days | Inhibition of proliferation | [20] |
| aluminiumneohydroxyaspergillin | Co-culture of *Aspergillus sclerotiorum* and *Penicillium* | U937 | IC50 4.2 µM; 48 h | Induction of cytotoxicity | [21] |
| Aranorosin | *Gymnascella dankaliensis* | L5178Y | IC50 0.58 µM | Induction of cytotoxicity | [22] |
| Aranorosin-2-methylether | *Gymnascella dankaliensis* | L5178Y | IC50 0.44 µM | Induction of cytotoxicity | [22] |
| Asteltoxin E | *Aspergillus carneus* | L5178Y | IC50 0.2 µM | Induction of cytotoxicity | [23] |
| Bromophilone B | *Penicillium canescens* | L5178Y | IC50 8.9 µM | Induction of cytotoxicity | [24] |
| Citriquinochroman | *Penicillium citrinum*, var *Cladosporium* | L5178Y | IC50 6.1 µM | Induction of cytotoxicity | [26] |
| Cladosporinone | *Cladosporioides (Fresen.)* G.A. de Vries | L5187Y | IC50 0.88 µM | Induction of cytotoxicity | [27] |
| Cyclo-(Gly-D-Leu-D-allo-Ile-L-Val-L-Val-D-Trp-β-Ala) | *Clonostachys rosea* | L5178Y | IC50 4.1 µM | Induction of cytotoxicity | [28] |
| Gymnastatin A | *Gymnascella dankaliensis* | L5178Y | IC50 0.64 µM | Induction of cytotoxicity | [22] |
| Gymnastatin B | | | IC50 5.80 µM | | [22] |
| Helvolic acid | *Aspergillus fumigatus* YK-7 | U937 | IC50 57.5 µM; 3 days | Inhibition of proliferation | [20] |
| Isopropylchetominine | *Aspergillus carneus* | L5178Y | IC50 0.4 µM | Induction of cytotoxicity | [23] |
| methyl 8-hydroxy-6-methyl-9-oxo-9H-xanthene-1-carboxylate | *Penicillium citrinum* var. | L5178Y | IC50 0.78, 1.0 µg/mL | Induction of cytotoxicity | [25] |
| Okaramine A | *Aphanoascus fulvescens (Cooke)* Apinis | L5178Y | IC50 4.0 µM | Induction of cytotoxicity | [29] |
| Okaramine C | | | IC50 12.8 µM | | |
| Okaramine G | | | IC50 13.8 µM | | |
| Okaramine H | | | IC50 14.7 µM | | |
| Pyripyropene E | *Aspergillus fumigatus* YK-7 | U937 | IC50 4.2 µM; 3 days | Inhibition of proliferation | [20] |
| Sterigmatocystin | *Aspergillus carneus* | L5178Y | IC50 0.3 µM | Induction of cytotoxicity | [23] |
| Verticillin D | *Clonostachys rosea* | L5178Y | IC50 0.1 µM | Induction of cytotoxicity | [28] |
| Viriditoxin | *Cladosporium* | L5187Y | IC50 0.1 µM | Inhibition of proliferation | [27] |

**Table 1.** *Cont.*

| Compound | Source | Cell Line/Animal Model | Dose; Duration | Efficacy | Reference |
|---|---|---|---|---|---|
| 1H-Dibenzo (b, e) (1, 4) dioxepin- 11-one,3, 8- dihydroxy- 4-(methoxymethyl)-1,6-dimethyl | *Lasiodiplodia theobromae* | L5178Y | IC50 7.3 μM | Induction of cytotoxicity | [30] |
| 4-Dehydroxy-altersolanol A | *Nigrospora oryzae* | L5178Y | IC50 9.4 μM | Induction of cytotoxicity | [31] |
| 5-methyl alternariol ether | *Penicillium citrinum* var. | L5178Y | IC50 0.78, 1.0 μg/mL | Induction of cytotoxicity | [25] |
| 9-Ethyliminomethyl-12-(morpholin- 4-ylmethoxy)-5,8,13,16-tetraaza -hexacene-2,3-dicarboxylic acid | cyanobacterium *Nostoc* sp. | DLA | IC50 372.4 ng/mL; 24 h | Induction of cytotoxicity | [32] |
| β-5,8,11-trihydroxybergamot-9-ene | *Aspergillus fumigatus* YK-7 | U937 | IC50 84.9 μM; 3 days | Inhibition of proliferation | [20] |

9-Ethyliminomethyl-12-(morpholin-4-ylmethoxy)-5, 8, 13, 16-tetraaza -hexacene-2, 3-dicarboxylic acid (EMTAHDCA).

*5.2. Plant-Derived Compounds and Lymphoma*

5.2.1. In Vitro Studies

There are several plant-derived compounds that showed an anticancer effect against lymphoma (Table 2). Muharini et al. reported that eleven of fifty-four compounds isolated from *Amorpha fruticosa* induced cytotoxicity against L5178Y mouse lymphoma cells [33]. As shown under the table, the IC50 values of eleven compounds were distributed between 0.2 and 10.2 µM, respectively. Baba et al. identified that apoptosis of primary effusion lymphoma (PEL) cells such as BC3, Ramos, BCBL1, and DG75 was induced under glucose-free conditions by arctigenin-a natural compound which has been studied for various bioactivity [34]. Arctigenin increased c-caspase-3, -9, and cleaved poly ADP ribose polymerase (c-PARP) in BC3, and triggered dissipation of mitochondrial membrane potential (MMP) together with decreased ATP production in BC3 and Ramos. Additionally, arctigenin induced up-regulation of transcriptional expressions of CHOP and ER stress-induced ER chaperon glucose related protein (GRP) 94, while down-regulating GRP78 and activating transcription factor 6$\alpha$ (ATF6$\alpha$) in PEL cells. Furthermore, arctigenin reduced the levels of p-p38 and p-extracellular signal-regulated kinase (ERK) 1/2, which play an important role in survival and apoptosis. Zhao et al. reported that the treatment of curcumin induced the antitumor effect on CH12F3 cell lines through DNA break and apoptosis [35]. This compound increased expression of nuclear $\gamma$H2AX, PAPR1, and proliferating cell nuclear antigen (PCNA), while decreasing Rad51 involved in homologous recombination which is essential for DNA repair. Additionally, up-regulation of caspase-3 and -9 was observed. These results showed that caspase-dependent apoptosis as well as DNA damage and impaired Rad51-dependent homologous recombination could be induced by curcumin. Tafuku et al. reported that natural carotenoid fucoxanthin (FX) and fucoxanthinol (FXOH) extracted from the brown seaweed *Cladosiphon okamuranus* Tokida showed an antitumor effect on BL and HL in vitro [36]. In many lymphoma cells, these compounds induced apoptosis (Raji, Daudi, B95-8/Ramos cell) and halted proliferation (Daudi, KM-H2, L540 cell) through cell cycle arrest at the G1 phase. In addition to the increase in c-PARP, c-caspase-3, and -9, the decrease in antiapoptotic and cell cycle regulatory proteins, including Bcl-2, cIAP-2, X-linked inhibitor of apoptosis protein (XIAP), cyclin D1, cyclin D2, was observed in Daudi after FXOH treatment. Additionally, the inhibition of nuclear factor $\kappa$B (NF$\kappa$B)-DNA binding activity was observed in Daudi by using electrophoretic mobility shift assay. These results indicated that FXOH induced apoptosis by decreasing NF$\kappa$B-DNA binding activity, followed by inhibition of antiapoptotic and cell cycle regulatory proteins. Mottaghipisheh et al. confirmed that various compounds isolated from *Ducrosia anethifolia* inhibited proliferation in L5178Y parent (PAR) and multidrug resistance (MDR) cells [37]. Among compounds, pabulenol, (þ)-oxypeucedanin hydrate, oxypeucedanin, oxipeudanin methanolated, imperatorin, isogospherol, heraclenin, and heraclenol had IC50 values for antiproliferation against L5178Y MDR cells, as well as L5178Y PAR cells. All values were indicated in the table and were less than 60.58 µM. Nakano et al. proposed the antiproliferative effects of MeOH extracts of aerial parts of *Citrus tachibana* against MT-1 and MT-2 cells [38]. Six phenanthroindolizidine alkaloids isolated from this extract showed antiproliferative activities at different EC50 values. Among the compounds, 3-demethyl-14a-hydroxyisotylocrebrine N-oxide showed toxicity in normal cells, while 3-demethyl-14b-hydroxyisotylocrebrine showed better potency than doxorubicin—a drug that is clinically used for antineoplasm. Methyl angolensate is a tetranortriterpenoid extracted from the root callus of *Soymida febrifuga,* called an Indian red wood tree [39]. Amounts of 10, 50, 100, 250 µM of methyl angolensate solution treatment for 24, 48, 72 h to Daudi cells caused the inhibition of cell proliferation, induction of apoptosis, ROS formation, and loss of mitochondrial transmembrane potential. These results suggest that methyl angolensate induces apoptosis through the mitochondrial pathway, supported by the induction of PARP cleavage. Further research revealed that methyl angolensate treatment facilitates DNA double-strand break repair by upregulation of nonhomologous DNA end joining (NHEJ) proteins including KU70 and KU80. These findings are in line with the upregulation of MRE11, RAD50, NBS1, and pATM, and downregulation of p53 and p73. Peperobtusin A extracted from *Peperomia tetraphylla* induced cell cycle arrest at the S phase and then apoptosis in U937 [40]. As for mechanisms, peperobtusin

A decreased MMP, Bid, caspase-3, and p38 while increasing ROS, c-caspase-8, -9, -3, and p-p38. Dissipation of MMP and ROS accumulation has been known to be involved in mitochondrial dependent apoptosis and cell death, respectively. Additionally, expression of Bcl-2, an antiapoptotic protein, was reduced, and Bax, a proapoptotic protein, was increased after peperobtusin A treatment. Taken together, the antitumor effect of peperobtusin A on U937 was induced by regulating the levels of pro-, antiapoptotic proteins, ROS, and MMP. Psilostachyin C, derived from *Ambrosia* spp. is among the types of sesquiterpene lactones [41]. Martino et al. reported that this natural compound (10 μg/ml for 24 h) showed significantly induced early and late apoptosis, necrosis, and cell cycle arrest at the S phase in murine lymphoma cell lines BW5147. Decreased levels of antioxidant enzymes such as superoxide (SOD), catalase (CAT), and peroxidase (Px) and up-regulation of rhodamine 123 negative cells were also reported. These results confirmed the efficacy of psilostachyin C to interfere with proliferation as well as mitochondrial functioning of lymphoma cells. Jara et al. reported that resveratrol exerted antiproliferative and proapoptotic effects on BL cell line Ramos [42]. Protein levels of apoptotic marker c-caspase-3 and c-PARP were increased. mRNA levels of proapoptotic mediators Noxa and p53 upregulated modulator of apoptosis (PUMA) also increased. In addition, resveratrol induced the up-regulation of primary double strand break sensor proteins as well as DNA damage and repair-related proteins including Rad50, Mre11, p-p95, p-ATM, p-BRCA1, γ-H2AX, DNA-PKcs, and KU80. These data suggested that resveratrol could regulate expression of genes with respect to DNA response, and then exert antiproliferative and apoptotic effects. Sui et al. adduced evidence that resveratrol regulates cell cycle arrest and apoptosis in extra nodal NK/T cell lymphoma (NKTCL) cell lines such as SNT-8, SNK-10, and SNT-16 [43]. After treatments, inconsistency for the percentage of cell cycle phase compared with untreated NKTCL cell lines was observed. Resveratrol induced an increase in the S phase with down-regulated expression of cyclin A2 which plays an important role in cell cycle progression. Decreased expression of Mcl-1 and survivin, and increased expression of Bax, Bad, c-caspase-3 and -9 were also observed. Phosphorylation of AKT and Stat3, known to be involved in proliferation, was inhibited by resveratrol. Furthermore, DNA damage response related proteins, including p-ATM, γ-H2A.X., p-chekpoint kinase (Chk) 2, and p-p53, were affected, and then increased. Schweinfurthins are natural compounds originated from *Macaranga alnifolia* Baker and are classified as members of the stilbene group [44]. Treatment of schweinfurthins at a dose of 100 nM against WSU-DLCL2, a phosphatase and tensin homolog (PTEN) deficient B cell lymphoma cells for 24 h increased the phosphorylation of eukaryotic initiation factor 2a (p-EIF2a) while decreasing the level of mTOR-AKT. Especially, schweinfurthin G elicited strong antiproliferative activities against three types of PTEN deficient B cell lymphoma cells—RL, SU-CHL-10, and WSU-DLCL2. These findings identified the potent inhibitory activities of this natural compound to treat PTEN deficient B cell lymphoma cells. Thymoquinone is a subtype of benzoquinone which was isolated from *Nigella sativa* Linn. Hussain et al. reported that thymoquinone (5 and 10 mM for 24 h) induced apoptosis and released ROS in activated B cell lymphoma cell lines (ABC-DLBCL) [45]. Nuclear compartments of HBL-1 and RIVA cell lines showed decreased translocation and phosphorylation of p65. Activation and cleavage of caspase-9, caspase-3, PARP, and Bax were examined, and inhibition of IκBa, XIAP, and survivin was also reported. It was also confirmed that thymoquinone treatment downregulated NFκB, and its transcriptional targets, Bcl-2 and Bcl-xL, subsequently decreased. Additionally, it was found that combination treatment of thymoquinone and TRAIL sufficiently inhibited cell viability and apoptosis, while each drug showed no effect when treated alone. These findings demonstrated that thymoquinone induced caspase and mitochondria dependent apoptosis against ABC-DLBCL. Hussain et al. administered thymoquinone to four primary effusion lymphoma cell lines—BC-1, BC-3, BCBL-1, and HBL-6, at the dose of 10 and 25 μM for 24 h [46]. Phosphorylation of AKT, forkhead box protein O1 (FOXO1), glycogen synthase kinase (GSK) 3, and Bad were suppressed in BC-1, BC-3, and BCBL-1 cells, suggesting that apoptosis was induced via the mitochondrial pathway. ROS release detected in BC-1 cells resulted in Bax activation and Bcl-2 inhibition. BC-1 and BC-3 cells showed cleavage of caspase-9, caspase-3, and PARP. Death receptor (DR) 5 upregulation was observed in BC-1 and BC-3 cells, but did not seem to be involved in thymoquinone-induced apoptosis. An alkaloid 4-Deoxyraputindole C extracted from *Raputia praetermissa*

induced cell death in Raji—a lymphoma cell line [47]. As for cell death, this compound decreased cell membrane integrity and increased lysosomal permeabilization. Additionally, dissipation of MMP and production of mitochondrial superoxide were observed by this compound. Cathepsins B/L, known as apoptosis and necrosis relating factors, were shown to be suppressed (IC 50 28.4 ± 1.2 µM and 1.7 ± 0.1 µM, respectively). Namely, 4-Deoxyraputindole C not only attenuates membrane integrity, MMP, and cathepsin but also increases the mitochondrial superoxide level. β-Asarone, known to show bioactivity such as anti-inflammation, revealed that it also induced antiproliferation and apoptosis in the lymphoma cell line Raji [48]. As a result of flow cytometry, an increase in the percentage of apoptotic cells after β-Asarone treatment was observed. The expression of c-caspase-3, -9, and c-PARP involved in intrinsic apoptosis was also up-regulated. Furthermore, decreased nuclear expression and phosphorylation of NFκB were confirmed, and TNFα-associated nuclear translocation of NFκB was reduced by treatment. Chen et al. demonstrated the mechanism of the anticancer activity of β-phenethyl isothiocyanate (PEITC) [49]. Administered to Raji cells, 10 µM PEITC increased cellular $H_2O_2$ levels and rapidly depleted cellular and mitochondrial glutathione. This led to the attenuation of mitochondrial respiration rate, following the disruption of mitochondrial respiratory complex I. NDUFS3, a mitochondrial respiratory complex I subunit, was especially the target of such disruption. Paterna et al. elucidated that dregamine and teberanaemontanine derivatives, which are indole alkaloids isolated from *Tabernaemontana elegans* Stapf. showed regulating effects against the proliferation of L5178Y cells [50]. Compounds 6, 8, 9, 10, 15, 16, and 23 induced cytotoxicity in significant IC50 levels, being ineffective in noncancerous mouse embryonic fibroblast NIH/3T3 cell line. Moreover, compounds 8, 9, 10, and 15 displayed strong MDR reversal activity, suggesting the potentiality of these derivatives to regulate lymphoma cell lines.

### 5.2.2. In Vitro and In Vivo Studies

Kumar et al. reported that chelerythrine, which originated from *Chelidonium majus*. L., is a rising antitumor agent given the previous reports proving its effects as a selective inhibitor of protein kinase C [51]. It was observed that survival duration was increased, and growth of Dalton's Lymphoma cells were reduced in BALB/c (H2d) mice when treated with this compound at 1.25 or 2.5 mg/kg for 34 days. It was also investigated that this compound (2.5 mg/kg) upregulated activating receptor NKG2D while downregulating inhibitory receptor NKG2A in TANK cells. These results supported the profound pharmaceutical activity of chelerythrine as a replacement for conventional therapies to treat lymphoma. Peters et al. demonstrated that elatol, a marine-derived natural compound, has effects of antiproliferation and apoptosis on lymphoma cell lines through translational inhibition of oncoproteins such as MYC, BCL-2, etc. [52]. It was confirmed that decreased proliferation and increased apoptosis were observed in SU-DHL-6, OCI-Ly3, and RIVA cells by elatol treatment. Of the cell lines, SU-DHL-6 and OCI-Ly3 were subject to further experiments. Results showed decrease in proteins expression, including cyclinD3, MYC, MCL1, PIM2, BCL2, and survivin, which were related in proliferation and apoptosis. Additionally, eIF4A activity which involved in translation was decreased by elatol. Furthermore, as results of in vivo experiment using xenograft mice, a reduction in tumor volume after elatol treatment for several weeks was observed. Yamamoto et al. reported that FX and FXOH induced apoptosis and cell cycle arrest in primary effusion lymphoma cells [53]. After treatments, increased levels of c-PARP, c-caspase-3, -8, and -9 were observed in BCBL-1 and TY-1. Following all experiments conducted on BCBL-1, decreased levels of antiapoptotic proteins Bcl-xL and XIAP were detected by both compounds. In addition, FXOH reduced survivin expression, AP-1 binding, and NFκB-DNA binding activity. pRb, known as cell cycle regulator, phosphorylation, and cell cycle regulatory proteins, including cyclin D2, cyclin dependent kinase (CDK) 4/6, and c-Myc, were decreased by treating both FX and FXOH. After treatments, the levels of p-IκB kinase (IKK) β, p-IkB α, IKK α, IKK β, IKK γ, Akt, phosphoinositide dependent protein kinase (PDK) 1, JunB, JunD, p-caspase-9, and β-catenin were also decreased. Moreover, FX treatment caused the loss of tumor weight in mice compared to untreated. These findings supported the potential of FX and FXOH for PEL therapeutic use. Yu et al. reported that cell cycle arrest at the G0/G1 phase and apoptosis were induced by pterostilbene (PTE) in mantle cell lymphoma cell lines [54]. In mantle

cell lymphoma cell lines Jeko-1, Granta-519, Mino, and Z-138, significant cytotoxicity was revealed, and percentage of apoptotic cells was also increased by PTE treatment. Further experiments were executed with Jeko-1 and Granta-519. While increasing expression of c-caspase-3, -8, -9, and Bax, PTE decreased expression of Bcl-2 and Bcl-xL. Additionally, PTE treatment induced dissipation of MMP and reduced expression of CDK4, CDK6, and cyclinD1, which are related in the cell cycle. Furthermore, a decrease in p-PI3K, p-Akt, p-mTOR, and p-p70S6K, which are involved in inactivation of the signaling pathway was observed. In the JeKo-1 xenograft model, expression of p-mTOR was decreased, and tumor growth was inhibited by PTE compared to the control treatment. Singh et al. studied the efficacy of resveratrol, a polyphenolic phytoalexin compound derived from various plants, including red grapes, mulberries, peanuts, and Japanese knotweed [55]. In this study, NOD/SCID mice were subcutaneously injected with EL4 cells. Five days post injection, they were orally treated with vehicle or resveratrol (10, 50, and 100 mg/kg) every day. Resveratrol suppressed EL4 tumor growth and increased survival time in a dose dependent manner. A following in vitro approach revealed that resveratrol (5, 10, 25, 50, 100 µM for 6, 12, 24 h) induced apoptosis of EL4 cells by elevating the expressions of AhR, Fas, FasL, Bax, Bid, cytochrome-c, sirtuin (SIRT) 1, cleaved caspase-8, -3, -9, and cleaved PARP and suppressing the phosphorylation of IκBα and expression of NF-κB. Xiao et al. verified that 11(13)-dehydroivaxillin (DHI) has antitumor effects on NHL cell lines, including Daudi, NAMALWA, SU-DHL-2, and SU-DHL-4, as well as xenograft mice [56]. To confirm the change in the percentage of apoptotic cells, flow cytometry analysis was used, and increased apoptosis was observed by DHI treatment in NHL cell lines. Cleavage of casapse-3 and PARP was shown in SU-DHL-2 and NAMLWA except Daudi and SU-DHL-4. The mRNA levels of IκB, cyclinD1, and Bcl-2 were decreased in Daudi, NAMALWA, and SU-DHL-2. The protein levels of p-IκBα, p-p65, IKKα/β, c-MYC, cyclinD1, and NF-κB were also reduced in Daudi and SU-DHL-2. Moreover, it was observed that DHI treatment induced the loss of tumor weight in experiments using an NHL xenograft mice model. As a result of immunohistochemistry (IHC), decreased proliferation markers, including PCNA, IKKα, and IKKβ were also confirmed. A number of plant-derived compounds were reported to have an effect on lymphoma cell lines, and they were studied focusing on cell cycle arrest, DNA repair, and the apoptosis pathway like other cancers. There were two studies on the effect of FXand FXOH extracted from *Cladosiphon okamuranus* Tokida on lymphoma cell line [36,53]. Both compounds showed cytotoxicity against lymphoma cell lines, including Daudi and BCBL-1, and upregulation of c-caspase-3, -8, -9, and c-PARP, which stand for apoptosis, was observed. Interestingly, studies taking both compounds reported that expressions of antiapoptotic proteins such as XIAP and Bcl-2 as well as NF-κB activity were reduced. It was also observed to induce cell cycle arrest at the G1 phase with decreased expression of the cell cycle regulator. In addition to in vivo study, an in vitro study using BCBL-1 xenograft model confirmed tumor weight loss after treatment with FXOH. These results could contribute to the development of an advanced drug for lymphoma. Psilostachyin C induced cell cycle arrest at the S phase, apoptosis, and necrosis in BW5147 cell lines [41]. A decrease in antioxidant enzymes and an increase in rhodamine 123 negative cells were also observed. Thus, research describes that this compound has both effects of antiproliferation and interfering mitochondrial function in lymphoma cell lines. As for ROS, methyl angolensate from *Soymida febrifuga* induced ROS generation, MMP dissociation, and apoptosis in Daudi cell lines. Additionally, proteins associated with NHEJ, a DNA repair mechanism, such as KU70 and KU80 were increased by this compound. Thymoquinone isolated from *Nigella sativa* was also observed by two studies. One reported the effect of this compound on activated BL and the other on PEL [45,46]. Both studies dealing with different types of lymphoma demonstrated that ROS generation was induced, followed by mitochondria mediated apoptosis by thymoquinone. Well known phytochemical resveratrol was also used in three different studies [42,43,55]. Resveratrol induced apoptosis of BL, NKTCL, murine lymphoma cell lines. Primary double strand sensor proteins and DNA damage and repair proteins, including Rad50, Mre11, p-ATM, p-BRACA1, DNA-PKcs, and KU80 were increased in BL cell lines by resveratrol. In NKTCL cell lines, DNA damage response related proteins, including p-ATM, γ-H2A.X., p-Chk2, and p-p53 were also affected by resveratrol. In addition to in vitro, it was observed that resveratrol suppressed EL4 tumor growth.

**Table 2.** Plant-derived compounds and lymphoma.

| System | Compound | Source | Cell Line/Animal Model | Dose; Duration | Efficacy | Mechanism | Reference |
|---|---|---|---|---|---|---|---|
| In vitro | Amorphispironones B | *Amorpha fruticosa* | L5178Y | IC50 7.6 µM | Induction of cytotoxicity | | [33] |
| In vitro | Arctigenin | | BC3 | Glucose(-), 5 µM; 2, 4, 6 h | Induction of apoptosis | ↑ c-caspase-3, -9, c-PARP ↓ MMP, ATP ↑ GRP94, CHOP ↓ p-p38, p-ERK1/2, p-p90RSK, GRP78, ATF6α ↑ GRP94, CHOP ↓ GRP78, ATF6α | [34] |
| | | | BC3, Ramos | Glucose(-), 1, 5, 10 µM | | | |
| | | | BC3, BCBL1 | Glucose(-), 5, 10 µM; 3, 4, 6 h | | | |
| | | | Ramos, DG75 | Glucose(-), 5 µM, 4, 6 h | | | |
| In vitro | Curcumin | | CH12F3 | 3, 6, 9, 10, 20, 30, 40, 50 µM; 4, 24 h | Induction of DNA damage and apoptosis | ↑ γH2AX, PAPR1, PCNA, caspase-3, -9 ↓ Rad51 | [35] |
| In vitro | Dalbinol | *Amorpha fruticosa* | L5178Y | IC50 0.2 µM | Induction of cytotoxicity | | [33] |
| | Dalpanol | *Amorpha fruticosa* | L5178Y | IC50 0.7 µM | Induction of cytotoxicity | | [33] |
| | Deguelin | *Amorpha fruticosa* | L5178Y | IC50 0.2 µM | Induction of cytotoxicity | | [33] |
| In vitro | Fucoxanthin | *Cladosiphon okamuranus Tokida* | Raji, Daudi, B95-8/Ramos | 5 µM; 24 h | Induction of apoptosis | | [36] |
| | | | Daudi, KM-H2, L540 | 2.5 µM; 24 h | Induction of cell cycle arrest | | |
| In vitro | Fucoxanthinol | *Cladosiphon okamuranus* Tokida | Raji, Daudi, B95-8/Ramos | 2.5 µM; 24 h | Induction of apoptosis | | [36] |
| | | | Daudi, KM-H2, L540 | 1.25 µM; 24 h | Induction of cell cycle arrest | | |
| | | | Daudi | 0.63, 1.25, 2.5, 5 µM; 24 h | Induction of apoptosis and cell cycle arrest | ↑ c-PARP, c-caspase-3, -9 ↓ Bcl-2, cIAP-2, XIAP, cyclin D1, cyclin D2, NF-κB-DNA binding | |

**Table 2.** *Cont.*

| System | Compound | Source | Cell Line/Animal Model | Dose; Duration | Efficacy | Mechanism | Reference |
|---|---|---|---|---|---|---|---|
| In vitro | Fuxocanthinol | *Cladosiphon okamuranus Tokida* | BCBL-1, TY-1 | 1.3, 2.5, 5 μM; 24 h | Induction of apoptosis | ↑ c-PARP, c-caspase-3, -8, -9 ↓ Bcl-xL, XIAP, survivin, p-pRb, cyclin D2, CDK4, CDK6, c-Myc, p-IKK β, p-IkB α, IKK α, IKK β, IKK γ, Akt, PDK1, p-cas9, β-catenin, JunB, JunD, NF-κB-DNA binding activity, AP-1 binding | [53] |
| | | | BCBL-1 | 1.3, 2.5, 5 μM; 24 h | Induction of cell cycle arrest | | |
| In vitro | Heraclenin | *Ducrosia anethifolia* | L5178Y PAR L5178Y MDR | IC50 32.73 μM IC50 46.54 μM | Inhibition of proliferation | | [37] |
| | Heraclenol | *Ducrosia anethifolia* | L5178Y PAR L5178Y MDR | IC50 52.31 μM IC50 46.57 μM | Inhibition of proliferation | | |
| In vitro | Hydroxyamorphispironone | *Amorpha fruticosa* | L5178Y | IC50 1.3 μM | Induction of cytotoxicity | | [33] |
| In vitro | Imperatorin | *Ducrosia anethifolia* | L5178Y PAR L5178Y MDR | IC50 36.12 μM IC50 42.24 μM | Inhibition of proliferation | | [37] |
| | Isogospherol | *Ducrosia anethifolia* | L5178Y PAR L5178Y MDR | IC50 46.53 μM IC50 48.75 μM | Inhibition of proliferation | | |
| In vitro | Isotylocrebrine | *Citrus tachibana* (Makino) T. Tanaka | MT-1 MT-2 MT-2 | EC50 48.3 nM; 4 h EC50 25.4 nM; 4 h EC50 13.0 nM; 4 h | Inhibition of proliferation | | [38] |
| In vitro | Isotylocrebrine Noxide | *Citrus tachibana* (Makino) T. Tanaka | MT-1 MT-2 | EC50 379.5 nM; 4 h EC50 246.7 nM; 4 h | | | [38] |
| In vitro | Methyl angolensate | *Soymida febrifuga* | Daudi | 10, 50 ,100, 250 μM; 24, 48, 72 h | Inhibition of proliferation Activation of apoptosis ROS formation | ↑ c-PARP, MRE11, RAD50, NBS1, p-ATM, KU70, KU80 ↓ p53, p73 | [39] |

**Table 2.** *Cont.*

| System | Compound | Source | Cell Line/Animal Model | Dose; Duration | Efficacy | Mechanism | Reference |
|--------|----------|--------|------------------------|----------------|----------|-----------|-----------|
| In vitro | Oxypeucedanin | *Ducrosia anethifolia* | L5178Y PAR L5178Y MDR | IC50 25.98 μM IC50 28.89 μM | Inhibition of proliferation | | [37] |
| | Oxypeucedanin methanolate | *Ducrosia anethifolia* | L5178Y MDR L5178Y PAR | IC50 35.88 μM IC50 33.23 μM | Inhibition of proliferation | | |
| | Pabulenol | *Ducrosia anethifolia* | L5178Y MDR L5178Y PAR | IC50 30.47 μM IC50 29.28 μM | Inhibition of proliferation | | |
| In vitro | Peperobtusin A | *Peperomia tetraphylla* | U937 | 25, 50, 75, 100 μM; 1, 3, 6, 24 | Induction of cell cycle arrest and apoptosis | ↑ ROS, Bax, c-caspase-8, -9, -3, p-p38 ↓ MMP, Bcl-2, Bid, caspase-3, p38 | [40] |
| In vitro | Psilostachyin C | *Ambrosia* spp. | BW5147 | 10 μg/mL; 24 h | Induction of apoptosis, necrosis Cell arrest in S phage Inhibition of cell viability, cell proliferation | ↓ SOD, CAT, Px | [41] |
| In vitro | Resveratrol | Various plants | Ramos | 20, 50, 70, 100 μM; 1, 3, 6, 10, 24 h | Induction of antiproliferative and proapoptotic activity | ↑ c-caspase-3, c-PARP, NOXA, PUMA, p-ATM, p-BRCA1, γ-H2AX, Rad 50, Mre 11, p-p95, DNA-PKcs, KU80 ↓ TCL-1. Myc, Bach2 | [42] |
| In vitro | Resveratrol | Various plants | SNT-8, SNK-10, SNT-16 | 25 μM; 0.5, 1, 3, 6, 12, 24, 48 h | Induction of cell cycle arrest Induction of DNA damage response and apoptosis Inhibition of proliferation | ↓ Cyclin A2 ↑ pATM, γ-H2A.X., p-Chk2, p-p53, Bax, Bad, c-caspase-9, -3 ↓ Mcl-1, survivin, p-AKT, p-Stat3 | [43] |

**Table 2.** *Cont.*

| System | Compound | Source | Cell Line/Animal Model | Dose; Duration | Efficacy | Mechanism | Reference |
|---|---|---|---|---|---|---|---|
| In vitro | Rotenone | *Amorpha fruticosa* | L5178Y | IC50 0.3 μM | Induction of cytotoxicity | | [33] |
| | rot-2′-enonic acid | *Amorpha fruticosa* | L5178Y | IC50 0.6 μM | Induction of cytotoxicity | | [33] |
| In vitro | Schweinfurthin | *Macaranga alnifolia* Baker | WSU-DLCL2 | 100 nM; 24 h | Inhibition of proliferation | ↑ p-EIF2a ↓ mTOR, AKT | [44] |
| In vitro | Sermundone | *Amorpha fruticosa* | L5178Y | IC50 0.2 μM | Induction of cytotoxicity | | [33] |
| In vitro | Thymoquinone | *Nigella sativa* | BC-1 | 10, 25 μM; 24 h | Induction of apoptosis Increase ROS generation Loss of MMP | ↑ Bax, c-caspase-3, -9, c-PARP, DR5 ↓ Bcl-2, p-AKT, p-FOXO1, p-GSK3, p-Bad | [46] |
| | | | BC-3 | | Induction of apoptosis, ROS generation | ↑ Bax, c-caspase-3, -9, c-PARP ↓ Bcl-2, p-AKT, p-FOXO1, p-GSK3, p-Bad | |
| | | | BCBL-1 | | Induction of apoptosis, ROS generation | ↓ p-AKT, p-FOXO1, p-GSK3, p-Bad | |
| | | | HBL-6 | | Induction of apoptosis | | |
| In vitro | Thymoquinone | *Nigella sativa* Linn. | ABC-DLBCL (HBL-1, RIVA) | 5, 10 mM; 24 h | Induction of ROS and apoptosis Inhibition of cell viability | ↑ c-caspase-9, -3, PARP, Bax ↓ NF-κB, IκBa, Bcl-2, Bcl-Xl, XIAP, Survivin, translocation of p65 subunit of NF-κB, p-p65 | [45] |
| In vitro | Tylophorine N-oxide | *Citrus tachibana* (Makino) T. Tanaka | MT-1 MT-2 | EC50 1590.0 nM; 4 h EC50 1490.0 nM; 4 h | Inhibition of proliferation | | [38] |
| In vitro | Tylophorinine N-oxide | *Citrus tachibana* (Makino) T. Tanaka | MT-1 MT-2 | (1)EC50 28.8 nM; 4 h (2)EC50 4.8 nM; 4 h | Inhibition of proliferation | | [38] |
| In vitro | 3-demethyl-14b-hydroxyisotylocrebrine | *Citrus tachibana* (Makino) T. Tanaka | MT-1 MT-2 | (1)EC50 2.8 nM; 4 h (2)EC50 2.6 nM; 4 h | Inhibition of proliferation | | [38] |

**Table 2.** *Cont.*

| System | Compound | Source | Cell Line/Animal Model | Dose; Duration | Efficacy | Mechanism | Reference |
|---|---|---|---|---|---|---|---|
| In vitro | 3, 3', 4-tri-O-methylellagic acid | *Combretum dolichopetalum* | L5179Y | IC50 29.0 μM | Induction of cytotoxicity | | [31] |
| In vitro | 4-Deoxyraputindole C | *Raputia praetermissa* | Raji | 20, 40, 60, 80, 100 μM; 6, 12, 24 h | Induction of cell death | ↑ mitochondrial superoxide ↓ MMP, cathepsin B/L | [47] |
| In vitro | 6a,12a- dehydrodeguelin | *Amorpha fruticosa* | L5178Y | IC50 10.2 μM | Induction of cytotoxicity | | [33] |
| | 6'-O-β-D -Glucopyranosyldalpanol | *Amorpha fruticosa* | L5178Y | IC50 1.7 μM | Induction of cytotoxicity | | [33] |
| In vitro | 14-hydroxytylophorine N-oxide | *Citrus tachibana* (Makino) T. Tanaka | MT-1 MT-2 | EC50 69.8 nM; 4 h EC50 26.8 nM; 4 h | Inhibition of proliferation | | [38] |
| In vitro | α-toxicarol | *Amorpha fruticosa* | L5178Y | IC50 0.2 μM | Induction of cytotoxicity | | [33] |
| In vitro | β-Asarone | | Raji | 100, 200, 400 μM; 72 h | Induction of apoptosis | ↑ c-caspase-9, -3, c-PARP ↓ procaspase-9, -3, PARP | [48] |
| | | | | 100 μM | Induction of anticancer effects | ↓ NF-κB/p65, p-NF-κB/p65, NF-κB/p65 nuclear translocation | |
| In vitro | β-Phenethyl isothiocyanate (PEITC) | | Raji | 10 μM; 3 h | Reduction in mitochondrial respiration rate Increase in cellular $H_2O_2$ levels Rapid depletion of cellular and mitochondrial glutathione | ↓ NDUFS3 | [49] |
| In vitro | (þ)-Oxypeucedanin hydrate | *Ducrosia anethifolia* | L5178Y MDR L5178Y PAR | IC50 41.96 μM IC50 60.58 μM | Inhibition of proliferation | | [37] |

**Table 2.** *Cont.*

| System | Compound | Source | Cell Line/Animal Model | Dose; Duration | Efficacy | Mechanism | Reference |
|---|---|---|---|---|---|---|---|
| In vitro | Compound 6<br>Compound 8<br>Compound 9<br>Compound 10<br>Compound 15<br>Compound 16<br>Compound 23 | *Tabernaemontana elegans* Stapf | L5178Y<br>L5178Y<br>L5178Y<br>L5178Y<br>L5178Y<br>L5178Y<br>L5178Y | IC50 11.38 μM; 24 h<br>IC50 63.91 μM; 24 h<br>IC50 35.56 μM; 24 h<br>IC50 29.21 μM; 24 h<br>IC50 34.28 μM; 24 h<br>IC50 20.77 μM; 24 h<br>IC50 33.30 μM; 24 h | Induction of cytotoxicity | | [50] |
| In vitro and in vivo | Chelerythrine | *Chelidonium majus.* L. | BALB/c (H2d) mice | 1.25, 2.5 mg/kg; 34 d | Increase in survival duration<br>Inhibition of Dalton's Lymphoma cell growth | | [51] |
| In vitro and in vivo | Elatol | | TANK | 2.5 mg/kg | | ↑ NKG2D<br>↓ NKG2A | [52] |
| | | | SU-DHL-6, OCI-Ly3, RIVA | 500 nM, 1, 10 μM; 24, 48, 72, 96 h | Induction of apoptosis | | |
| | | | SU-DHL-6, OCI-Ly3 | 5 μM; 16 h | | ↓ cyclinD3, MYC, MCL1, PIM2 | |
| | | | | 1, 10 μM; 4, 16, 24 h | Inhibition of protein synthesis | | |
| | | | OCI-Ly3 | 5 μM; 16 h | | ↓ BCL-2, survivin | |
| | | | SCID mice(engrafted with SU-DHL-6) | 20 mg/kg; 20 days | Inhibition of tumor growth | | |
| | | | SCID mice(OCI-Ly3 xenograft) | 40 mg/kg; 30 days↑ | | | |
| In vitro and in vivo | Fucoxanthin | *Cladosiphon okamuranus* Tokida | BCBL-1, TY-1 | 2.5, 5, 10 μM; 24 h | Induction of apoptosis | ↑ c-PARP, c-caspase-3. -8, -9 ↓ Bcl-xL, XIAP, p-pRb p-IKK β, p-IkB α, IKK α, IKK β, IKK γ, Akt, PDK1, p-caspase-9, β-catenin, JunB, JunD | [53] |
| | | | BCBL-1 | 2.5, 5, 10 μM; 24 h | Induction of cell cycle arrest | | |
| | | | SCID mice (BCBL-1 Xenograft) | 150 mg/kg; 56 days | Inhibition of tumor growth | | |

**Table 2.** *Cont*.

| System | Compound | Source | Cell Line/Animal Model | Dose; Duration | Efficacy | Mechanism | Reference |
|---|---|---|---|---|---|---|---|
| In vitro and in vivo | Pterostilbene | | Jeko-1,Granta-519, Mino, Z-138 | 10, 20, 40, 60, 80 µM; 24, 48, 72 h | Induction of cytotoxicity | ↑ c-caspase-3, -8, -9, Bax ↓ CDK4, CDK6, cyclinD1, MMP, Bcl-2, Bcl-xL, p-PI3K, p-Akt, p-mTOR, p-p70S6K | [54] |
| | | | Jeko-1, Granta-519 | 20, 40, 80 µM; 48 h | Induction of cell cycle arrest and apoptosis | | |
| | | | NOD/SCID mice (JeKo-1 Xenograft) | 50 mg/kg; 15 days | Inhibition of tumor growth | ↓ p-mTOR | |
| In vitro and in vivo | Resveratrol | Various plants | EL4 | 5, 10, 25, 50, 100 µM; 6, 12, 24 h | Induction of apoptosis | ↑ AhR, Fas, FasL, Sirt1, Bax, Bid, cytochrome-c, SIRT1 c-caspase-8, -3, -9, c-PARP ↓ p-IκBα, NF-κB | [55] |
| | | | NOD/SCID/$\gamma$c$^{null}$ mice/EL4 | 10, 50, 100 mg/kg; 38 days | Suppression of tumor growth Increase in survival time | | |
| In vitro and in vivo | 11(13)-dehydroivaxillin | *Carpesium genus* | Daudi, Namalwa, SU-DHL-4, SU-DHL-2 SU-DHL-2, NAMALWA | 5, 7, 10 µM; 24 h | Induction of apoptosis | ↑ c-PARP, c-caspase-3 | [56] |

**Table 2.** *Cont.*

| System | Compound | Source | Cell Line/Animal Model | Dose; Duration | Efficacy | Mechanism | Reference |
|---|---|---|---|---|---|---|---|
| | | | Daudi, NAMALWA, SU-DHL-2 | 5, 10 µM; 6 h | | ↓ cyclin D1, Bcl-2, IκBα, | |
| | | | Daudi, SU-DHL-2 | 10 µM; 4 h | | ↓ p-IκBα, p-p65 | |
| | | | | 5, 7 µM; 24 h | | ↓ IKKα/IKKβ, c-MYC, cyclinD1, NF-κB | |
| | | | B-NSG mice(Daudi, SU-DHL-2 xenograft) | 50 mg/kg; 10 days | Inhibition of tumor growth | ↓ IKKα/IKKβ, PCNA | |

Cleaved poly ADP ribose polymerase (c-PARP); cleaved caspase (c-caspase); procaspase (p-caspase); reactive oxygen species (ROS); superoxide dismutase (SOD); catalase (CAT); peroxidase (Px); death receptor (DR); mitochondrial membrane potential (MMP); X-linked inhibitor of apoptosis protein (XIAP); B-cell lymphoma-extra large (Bcl-xL); retinoblastoma protein (Rb); IκB kinase (IKK); phosphoinositide dependent protein kinase (PDK); nuclear factor kappa B (NFκB); activator protein (AP); cellular inhibitor of apoptosis protein (cIAP); forkhead box protein O1 (FOXO1); glycogen synthase kinase (GSK); proliferating cell nuclear antigen (PCNA); glucose-regulated protein (GRP); CCAAT/enhancer-binding protein homologous protein (CHOP); extracellular signal-regulated kinase (ERK); activating transcription factor (ATF); cyclin dependent kinase (CDK); phosphatidylinositol 3-kinase (PI3K); sirtuin (SIRT); DNA dependent protein kinase (DNA PK); signal transducer and activator of transcription (STAT); p53 upregulated modulator of apoptosis (PUMA); checkpoint kinase (Chk); death receptor (DR).

### 5.3. Animal-Derived Compounds and Lymphoma

Some animal-derived compounds were reported to have anticancer efficacies against lymphoma (Table 3). Annomontine, ingenines A, and B are alkaloids obtained from the methanol extract of *Acanthostrongylophora ingens,* the Indonesian sponge [57]. Annomontine and ingenines B exerted cytotoxicity against L5178Y cells with the ED50 values 7.8 and 9.1 μg/mL, respectively. However, ingenines A showed little cytotoxic properties. Mokhlesi et al. demonstrated that isolated compounds from Indonesian marine sponge of the genus *Cinachyrella* exerted cytotoxicity against L5178Y mouse lymphoma cells [58]. This sponge is known for its therapeutic effects to produce steroids and fatty acid metabolisms. Cinachylenic acids A, B, C, and D were acetylenic acid derivatives exhibiting pronounced cytotoxicity with an IC50 value of 0.3 μM. Cinobufotalin is a kind of bufadienolide which originates from the venom of the skin secretions of giant toads such as *Bufo gargarizans* [59]. Cell viability and MMP declined in the U937 cells treated with cinobufotalin (0.5 and 1 μM for 6, 12 and 24 h). Rapid release of cytosolic superoxide anion and increase in intracellular $Ca^{2+}$ were also observed. Expressions of Fas and caspase-3 and caspase-8 cleavage were elevated, whereas procaspase-2, -3, -8, -9, cytosolic Bid, and Bax protein levels declined. These findings suggest that cinobufotalin induces apoptosis via both the intrinsic and extrinsic pathways. Dyshlovoy et al. reported that frondoside A (FrA), a natural compound extracted from sea cucumber *Cucumaria okhotensis*, exerted caspase-/p53-independent apoptosis and inhibited prosurvival autophagy in BL cells [60]. After FrA treatment, cell cycle arrest at the G1 phase was observed in CA46, Namalwa, Ramos, and BL-2. Cleavage of PARP and decreased survivin and Bcl-2, which are known as antiapoptotic proteins, were observed in CA46. It was also confirmed that FrA induced translocation of mitochondrial proteins, including cyt C, apoptosis inducing factor (AIF), and HtrA1/Omi, from mitochondria to cytosol in CA46, BL-2, and Ramos. In addition, accumulation of LC3B-I/II and sequestosome (SQSTM) 1/p62 known to be associated with autophagy was verified. Iodocionin is a compound isolated from the mediterranean ascidian *Ciona edwardsii* by A. Aiello et al. [61]. L5178Y mouse lymphoma cell lines treated with 0.1, 0.3, 1, 3, 10 μg/mL idiocionin formula for 72 h displayed lower levels of cell viability. The brominated analogue of idiocionin, its iodine transferred into bromine, featured similar bioactivity against L5178Y cells. Ebada et al. reported that compounds isolated from marine sponge *Jaspis splendens* such as (+)-jasplakinolide Z6, (+)-jasplakinolide, (+)-jasplakinolide Z5, and (+)-jasplakinolide V exhibited cytotoxicity against L5178Y [62]. The IC50 value of (+)-jasplakinolide Z6 was 3.2 μM, and the rest was less than 100 nM. Interestingly, frondoside A(frA), extracted from sea cucumber *Cucumaria okhotensis*, changed not only autophagy relating proteins such as LC3B-I/II but apoptosis relating proteins in BL cells [60]. Cinobufotalin, known to one of toad venom components has been reported to have anticancer effects in various cancer cell lines including esophageal squamous cell carcinoma and the melanoma cell line [63,64]. This anticancer effect was also observed in U937-a lymphoma cell line [59]. Cinobufotalin induced changes in intracellular $Ca^{2+}$, MMP and expression of apoptosis relating proteins such as Fas, caspase, Bid, and Bax. Marin sponge derived compounds, (+)-jasplakinolide Z6, (+)-jasplakinolide, (+)-jasplakinolide Z5, and (+)-jasplakinolide V, showed cytotoxicity against L5178Y [62]. These compounds exhibited prominent IC50 values of less than 100 nM except for (+)-jasplakinolide Z6.

**Table 3.** Animal-derived compounds and lymphoma.

| Compound | Source | Cell Line/Animal Model | Dose; Duration | Efficacy | Mechanism | Reference |
|---|---|---|---|---|---|---|
| Annomontine | *Acanthostrongylophoraingens* | L5178Y | ED50 7.8 µg/mL | Induction of cytotoxicity | | [57] |
| Cinachylenic Acid A, B, C, D | *Cinachyrella* sp. | L5178Y | IC50 0.3 µM | Induction of cytotoxicity | | [58] |
| Cinobufotalin | Toad | U937 | 0.5, 1 µM; 6, 12, 24 h | Decrease in cell viability and MMP Rapid release of cytosolic superoxide anion, increase in intracellular $[Ca^{2+}]$ | ↑ Fas, c-caspase-3, -8 ↓ Pro-caspase-2, -3, -8, -9, cytosolic Bid, cytosolic Bax | [59] |
| Frondoside A | *Cucumaria frondosa* | CA46, Namalwa, Ramos, BL-2 | 0.3, 0.6 µM; 48 h | Induction of cell cycle arrest | | [60] |
| | | CA46 | 0.3 µM; 48 h | Inhibition of prosurvival autophagy | ↑ LC3B-I/II, SQSTM1/p62 | |
| | | | 0.3, 0.6 µM; 48 h | Induction of apoptosis | ↑ c-PARP ↓ Survivin, Bcl-2 | |
| | | CA46, BL-2, Ramos | 0.3, 0.6 µM ;48 h | Induction of apoptosis | ↑ Cyt C, AIF, HtrA2/Omi | |
| Ingenine B | *Acanthostrongylophoraingens* | L5178Y | ED50 9.1 µg/mL | Induction of cytotoxicity | | [57] |
| Iodocionin | *Ciona edwardsii* | L5178Y | 0.1, 0.3, 1, 3, 10 µg/mL; 72 h | Inhibition of cell proliferation | | [61] |
| (+)-Jasplakinolide. (+)-Jasplakinolide Z5, (+)-Jasplakinolide V (+)-Jasplakinolide Z6 | *Jaspis splendens* | L5178Y | IC50 < 100 nM  IC50 3.2 µM | Induction of cytotoxicity | | [62] |

Cleaved poly ADP ribose polymerase (c-PARP); sequestosome (SQSTM); cytochrome C (Cyt C); apoptosis inducing factor (AIF).

## 6. Discussion

Lymphoma, a kind of blood cancer can be classified into HL (10%) and NHL (90%) [3]. There is no clear etiology, but it has been reported that heredity, environment, and infection such as Ebstein Barr Virus (EBV), *Helicobactor pylori* and *Chlamydia psitacci* might be responsible for pathogenesis [65]. Among subtypes, it is known that B cell-associated lymphoma is the most common lymphoma. In view of molecular biology, aberrant B cell receptor (BCR) signaling contributes to B cell abnormality. Activated BCR leads to LYN kinase mediated cascade, and succeeding proteins such as SYK, BTK, and the CARD11-BCL10-MALT1 complex were activated. Subsequently, NF-κB free from IKK induces proliferative and antiapoptotic gene expression [5,66,67]. The schematic diagrams of cell cycle- and apoptosis-related factors are elucidated in Figures 1 and 2, respectively. These have been targeted for the development of future drugs for lymphoma. In agreement with expectations, anticancerous effects of various substances were confirmed under experimental conditions. However, there are blanks of mechanisms in many compounds not just microorganism-derived compounds. To see how compounds affect cell viability, further study for mechanism seems to be required. Additionally, it seems be better to test cytotoxicity against normal cell lines as natural compounds might have an adverse effect.

In the current condition of growing interest in natural products to minimize the adverse effects of conventional treatment, we explored natural compounds for lymphoma and sorted them into each table by their source. It was confirmed that various compounds showed remarkable effects on lymphoma under in vitro and in vivo experiments. These systemic data will provide researchers with inspiration for the development of novel drugs for lymphoma and contribute to future studies.

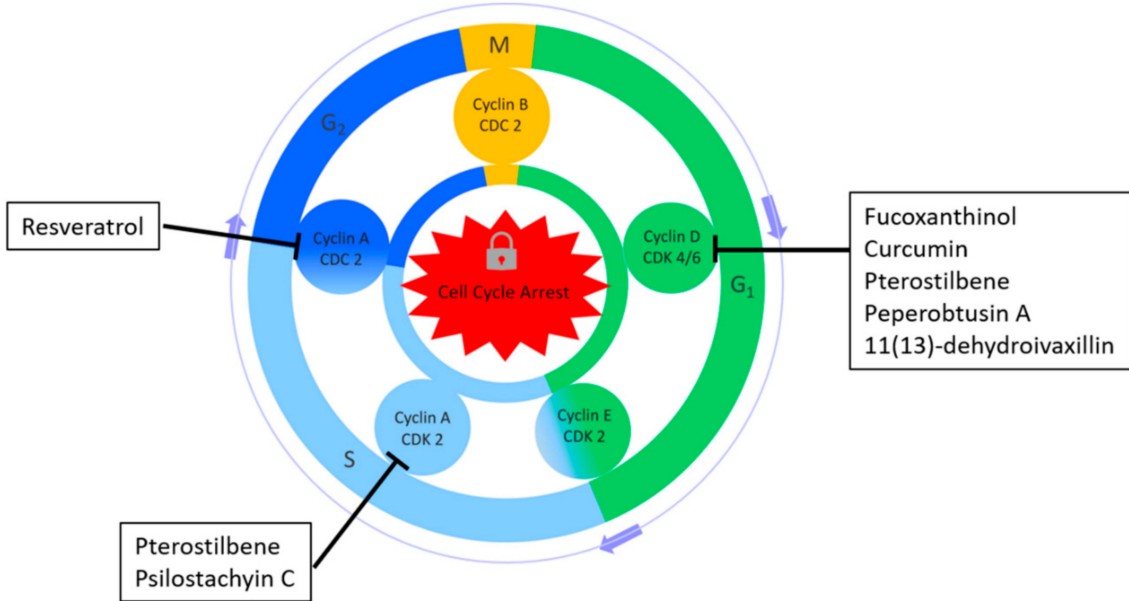

**Figure 1.** Schematic diagram of natural product-induced cell cycle regulation.

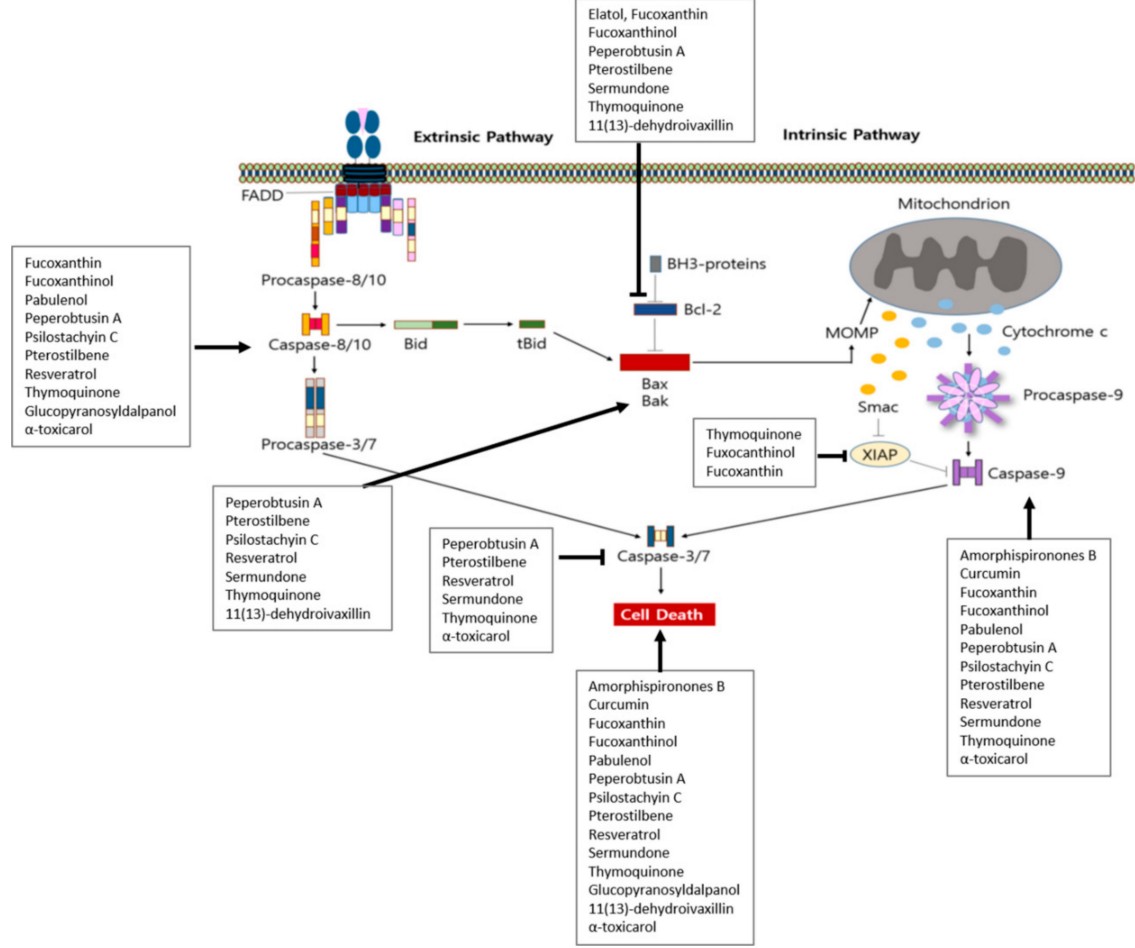

**Figure 2.** Schematic diagram of natural product-induced apoptosis.

*Limitation of This Study*

Although our study provides researchers with systemic data about natural products for lymphoma, it also has drawbacks. First, natural products as well as decoctions were known to have a synergistic effect by inner compound to compound cooperation, but we almost categorized natural compounds. Second, no clinical trials were included in this review. Third, as new studies are incessantly being reported, there is a term limit lasting 10 years as indicated in the Methods paragraph.

## 7. Conclusions

In this study, anticancerous abilities of ninety natural compounds against lymphoma were reviewed. Each natural compound regulated diverse mechanisms including antiproliferation, apoptosis, and cell cycle arrest. These have a potential to treat diverse types of lymphoma. If further studies were involved from now on, natural compounds may become a promising strategy for lymphoma instead of conventional treatment.

**Author Contributions:** Conceptualization, Y.C. and B.K.; methodology, Y.C.; investigation, Y.C., S.N., and S.Y.K.; data curation, Y.C. and B.K.; writing-original draft preparation, Y.C., M.N.P., S.N., and S.Y.K.; writing-review and editing, B.K.; visualization, Y.C. and M.N.P.; supervision, M.N.P. and B.K.; project administration, B.K.; funding acquisition, B.K. All authors have read and agreed to the published version of the manuscript.

**Funding:** This work was supported by the National Research Foundation of Korea (NRF) grant funded by the Korea government (MSIT) (No. 2020R1A5A201941311) and Basic Science Research Program through the National Research Foundation of Korea (NRF) funded by the Ministry of Education (NRF-2020R1I1A2066868).

**Conflicts of Interest:** The authors declare no conflict of interest.

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
