# Peer review of "Review of Natural Compounds for the Management and Prevention of Lymphoma"

_processes, doi:10.3390/pr8091164_

Round 1
Reviewer 1 Report
The review is interesting and the authors present studies concerning the anti-cancerous effects on lymphoma cells by natural products.
This review should be considered for publication on MDPI after minor revision as listed below.
Section 4.2 entitled "Plant-derived compounds and lymphoma" is lengthy and is dispersive. Authors should divide section 4.2 for example in according to different two types of lymphoma, Hodgkin Lymphoma and non-Hodgkin Lymphoma, or different types of natural compounds or based on in vitro and in vivo studies.
In sections 4.1, 4.2 and 4.3, the authors should insert a schematic diagram with the mechanisms of action of the most interesting and promising natural compounds in lymphomas.
The authors show the data of anti-cancer effects in vitro and in vivo, it would be interesting to include clinical trials if they are present in the literature.
The authors should insert a section with combinatorial treatment with natural compounds in lymphoma.
Author Response
We appreciate editors and reviewers for critical comments to improve the quality of our manuscript (processes-916085), titled “Review of natural compounds for the management and prevention of lymphoma”. We earnestly responded to the raised comments point by points.
Section 4.2 entitled "Plant-derived compounds and lymphoma" is lengthy and is dispersive. Authors should divide section 4.2 for example in according to different two types of lymphoma, Hodgkin Lymphoma and non-Hodgkin Lymphoma, or different types of natural compounds or based on in vitro and in vivo studies.
(Response): Thanks. 4.2 is divided in according to its experimental model.
In sections 4.1, 4.2 and 4.3, the authors should insert a schematic diagram with the mechanisms of action of the most interesting and promising natural compounds in lymphomas.
(Response): Thanks, mechanisms of compounds were added in preexisting figures.
The authors show the data of anti-cancer effects in vitro and in vivo, it would be interesting to include clinical trials if they are present in the literature.
(Response): Thank you for the comments. However, clinical trials about natural products against lymphoma will be reviewed soon in another study.
The authors should insert a section with combinatorial treatment with natural compounds in lymphoma.
(Response): You are right. The combinatorial treatment with natural compounds in lymphoma is very important issue in oncology. For this study, we aimed to review anti-lymphomatic efficacies of natural compounds themselves. For the next ongoing study, we will review combinatorial treatments of natural compounds with conventional chemotherapies and clinical trials which contains more clinical evidences for lymphoma treatment.
Again we appreciate three reviewers and editors for their kind and careful comments for improving the quality of our manuscript and also sincerely hope we address our responses well to the raised comments and our revised manuscript would be accepted for publication in your journal soon.
With kind regards,
Prof. Bonglee Kim, M.D, Ph.D.
Department of Pathology, College of Korean Medicine, Kyung Hee University
1 Hoegi-dong, Dongdaemun-ku, Seoul 130 -701, South Korea
E-mail: bongleekim@khu.ac.kr
Tel; +82-2-961-9217, Fax; +82-2-961-9217
Reviewer 2 Report
The manuscript by the Authors is an extensive overview on the topic.
- I am not sure that a review article should comprise a discussion section and indeed it contains large overlaps with what written before. The Authors should merge this section with the previous parts adding the comments where appropriate.
- The overview on lymphomas in the discussion is very generic and superficial. It can be deleted, while a brief introduction on this type of cancer shuold be provided at the beginning of the manuscript to introduce the topic of the review.
- Figure 1 and 2 should highlight the proteins/processes targeted by the classes of natural compounds
Minor comments:
- “ABC cells, which are subtypes of DLBCL.”: The Authors should write ABC-DLBCL (also in other parts of the manuscript).
- The Authors should use the already introduced abbreviations.
Author Response
We appreciate editors and reviewers for critical comments to improve the quality of our manuscript (processes-916085), titled “Review of natural compounds for the management and prevention of lymphoma”. We earnestly responded to the raised comments point by points.
I am not sure that a review article should comprise a discussion section and indeed it contains large overlaps with what written before. The Authors should merge this section with the previous parts adding the comments where appropriate.
(Response): Thanks. The discussion section is merged with proper parts to avoid the overlaps.
The overview on lymphomas in the discussion is very generic and superficial. It can be deleted, while a brief introduction on this type of cancer shuold be provided at the beginning of the manuscript to introduce the topic of the review.
(Response): The overview of lymphoma is deleted. The brief introduction of lymphoma is added in “1. Lymphoma” section.
Figure 1 and 2 should highlight the proteins/processes targeted by the classes of natural compounds
(Response): Thanks. The targeted proteins/processes of natural compounds were added in figure 1 and 2.
Minor comments:
“ABC cells, which are subtypes of DLBCL.”: The Authors should write ABC-DLBCL (also in other parts of the manuscript).
(Response): Revised.
The Authors should use the already introduced abbreviations.
(Response): Revised.
Again we appreciate three reviewers and editors for their kind and careful comments for improving the quality of our manuscript and also sincerely hope we address our responses well to the raised comments and our revised manuscript would be accepted for publication in your journal soon.
With kind regards,
Prof. Bonglee Kim, M.D, Ph.D.
Department of Pathology, College of Korean Medicine, Kyung Hee University
1 Hoegi-dong, Dongdaemun-ku, Seoul 130 -701, South Korea
E-mail: bongleekim@khu.ac.kr
Tel; +82-2-961-9217, Fax; +82-2-961-9217
Round 2
Reviewer 2 Report
The manuscript has been improved. I still find some statements such as
"Lymphomas affect any organs in human body showing Though a wide range of symptoms including night sweat, fever, and weight loss, etc. were representative symptoms of lymphoma,and asymptomatic patients also were knownre ported. Diagnosis is conducted by lymph node biopsy, considering numerous factors that can affect condition of patient who suffering lymphoma"
and
"Especially, R-CHOP, a well-known lymphoma .... has reported to have adverse effects such as nausea, fatigue, hair loss and so on"
as not appropriate for a scientific publication. Perhaps, an oncologist might help also in rewriting the cancer section.
Why do the Authors refer to system biology?
Author Response
We appreciate reviewer for critical comments to improve the quality of our manuscript (processes-916085), titled “Review of natural compounds for the management and prevention of lymphoma”. We earnestly responded to the raised comments point by points.
"Lymphomas affect any organs in human body showing Though a wide range of symptoms including night sweat, fever, and weight loss, etc. were representative symptoms of lymphoma,and asymptomatic patients also were knownre ported. Diagnosis is conducted by lymph node biopsy, considering numerous factors that can affect condition of patient who suffering lymphoma"
and
"Especially, R-CHOP, a well-known lymphoma .... has reported to have adverse effects such as nausea, fatigue, hair loss and so on"
as not appropriate for a scientific publication. Perhaps, an oncologist might help also in rewriting the cancer section.
(Response): Thanks. The mentioned parts were rewrote with consultation with oncologist.
Why do the Authors refer to system biology?
(Response): As mentioned in manuscript, cancer is a complex ecosystem where heterogeneous tumor cells and non-tumor cells coexist. That makes uneasy to treat cancer because anti-cancer drugs have shown side effects and relapses problems. At this point, the systems biology approaches can be used for effective strategies. We hope that this article could be used as basic study for development of novel anti-lymphoma drugs in the aspect of cancer system biology through combination of the natural compounds with potent efficacies against lymphoma. That is the reason why we referred to the system biology.
Again we appreciate reviewer for your kind and careful comments for improving the quality of our manuscript and also sincerely hope we address our responses well to the raised comments and our revised manuscript would be accepted for publication in your journal soon.
With kind regards,
Prof. Bonglee Kim, M.D, Ph.D.
Department of Pathology, College of Korean Medicine, Kyung Hee University
1 Hoegi-dong, Dongdaemun-ku, Seoul 130 -701, South Korea
E-mail: bongleekim@khu.ac.kr
Tel; +82-2-961-9217, Fax; +82-2-961-9217